# Impact of Switching from Oral to Long-Acting Injectable Cabotegravir and Rilpivirine on the Lipid Profile of HIV-Positive Patients

**DOI:** 10.3390/microorganisms14010022

**Published:** 2025-12-21

**Authors:** Marta Segura Díaz, Antonio Collado Romacho, Sergio Ferra Murcia

**Affiliations:** 1Department of Internal Medicine, Hospital Universitario Torrecárdenas, 04009 Almería, Spain; marta_sep95@hotmail.com; 2Clinical Management Unit of Infectious Diseases, Hospital Universitario Torrecárdenas, 04009 Almería, Spain

**Keywords:** long-acting therapy, HIV, dyslipidemia, cabotegravir, rilpivirine, lipid profile, HDL

## Abstract

Long-acting cabotegravir and rilpivirine (LA-CAB/RPV) have been incorporated into the treatment of people living with HIV (PLWH), but evidence on their metabolic impact in real-world settings remains limited. This retrospective study analyzed the lipid profiles of 39 PLWH who switched from daily oral antiretroviral therapy to LA-CAB/RPV. Lipid parameters were compared before and seven months after the switch. No significant differences were observed in total cholesterol, LDL cholesterol, or triglycerides, indicating that LA-CAB/RPV did not worsen the lipid profile. However, HDL cholesterol increased significantly from 49.4 ± 11.5 mg/dL to 53.0 ± 11.9 mg/dL (*p* = 0.0065). Viral suppression and CD4 counts remained stable throughout the study period. These findings suggest that switching to long-acting injectable cabotegravir and rilpivirine maintains virological and immunological control without adversely affecting the total cholesterol, LDL cholesterol, or triglycerides, and is associated with an improvement in HDL cholesterol. LA-CAB/RPV therefore appears to be a metabolically safe therapeutic option for PLWH, with a potentially favorable effect on cardiovascular risk factors.

## 1. Introduction

The highly active antiretroviral therapy regimen has made it possible to drastically reduce mortality (by 45% since 2005) and complications related to immunosuppression caused by HIV in people infected with this virus. As a result, the life expectancy of people living with HIV (PLWH) has been increasing over the past decades, almost reaching that of the seronegative population of the same age and sex. In this regard, it is estimated that by 2030, possibly 70% of PLWH will be over 50 years old [1].

Cardiovascular disease is one of the leading causes of death in this population. Several large cohort studies and meta-analyses have shown that the cardiovascular risk in people living with HIV may be up to twice that of the general population [2,3,4]

This phenomenon could be related to several factors, such as an inflammatory state, chronic or persistent immune activation, and the chronic administration of antiretroviral therapy with known metabolic and mitochondrial toxicity, among others [5,6].

While viral suppression achieved through ART improves immune recovery and reduces systemic inflammation, long-term exposure to certain agents—particularly protease inhibitors and some nucleoside reverse transcriptase inhibitors—has been associated with metabolic and endothelial toxicities that contribute to cardiovascular risk. Recent cohort and mechanistic studies have shown that ART may exert both cardioprotective and potentially cardiotoxic effects, depending on the class and duration of exposure [4,7,8].

Additionally, type II diabetes mellitus, hypertension, dyslipidemia, and smoking are more prevalent in PLWH than in the general population [1].

Dyslipidemia refers to an abnormality in lipid metabolism that results in elevated concentrations of total cholesterol, low-density lipoprotein cholesterol (LDL-C), or triglycerides, and/or decreased concentrations of high-density lipoprotein cholesterol (HDL-C). Such alterations are recognized as important risk factors for the development of cardiovascular disease. Dyslipidemia is highly prevalent among PLWH and represents one of the major contributors to cardiovascular risk in this population [9].

Antiretroviral therapy causes alterations in lipid metabolism. The antiretrovirals most commonly associated with these changes have been protease inhibitors (PIs) and nucleoside reverse transcriptase inhibitors (NRTIs). Therefore, the patient’s metabolic profile and the prevention of cardiovascular events are important considerations when selecting antiretroviral therapy [1].

Recent evidence has highlighted the metabolic implications of newer antiretroviral drug classes, particularly integrase strand transfer inhibitors (INSTIs) and second-generation non-nucleoside reverse transcriptase inhibitors (NNRTIs). Second-generation INSTIs, such as dolutegravir and bictegravir, have been associated with clinically significant weight gain in people living with HIV (PLWH) [10]. Despite this weight increase, these agents generally exert a neutral or only modestly adverse effect on lipid parameters compared with older drug classes [11]. In contrast, second-generation NNRTIs such as rilpivirine have demonstrated a more favorable lipid profile than efavirenz, with lower increases in total cholesterol, LDL cholesterol, and triglycerides, and without a strong association with weight gain [12]. Given that the majority of our cohort was receiving INSTI- or rilpivirine-based regimens prior to switching to long-acting injectable therapy, these characteristics are relevant for contextualizing the metabolic outcomes observed in the present study.

Adherence to antiretroviral therapy is a key determinant of treatment success, ensuring viral suppression and preventing resistance development [13]. However, it can be affected by stigma, fear of disclosure, treatment fatigue, and regimen complexity. Long-acting injectable regimens, such as cabotegravir and rilpivirine, reduce the need for daily oral dosing and may therefore facilitate sustained adherence [14]. In January 2021, this combination was approved by the U.S. Food and Drug Administration (FDA) for monthly administration, followed in February 2022 by approval for dosing every two months [15].

For all of the reasons stated above, the primary objective of this study was to evaluate the impact on the lipid profile in PLWH who switched from daily oral therapy to long-acting injectable therapy with cabotegravir and rilpivirine. Secondary objectives included evaluating previous lines of antiretroviral therapy, mapping the magnitude of differences according to prior treatments received, describing the evolution of CD4 levels and viral load, and determining whether any deaths or cardiovascular events occurred following the switch to injectable therapy.

## 2. Materials and Methods

This was a descriptive, observational, longitudinal, and retrospective study that evaluated changes in lipid profile over a 7-month period in HIV-positive adults who switched from daily oral antiretroviral therapy to long-acting injectable cabotegravir plus rilpivirine, administered every two months.

### 2.1. Participants and Eligibility

Patients were eligible if they met the standard criteria for switching to LAT according to prescribing information and clinical guidelines, including sustained viral suppression for at least six months, absence of resistance to cabotegravir or rilpivirine, and no prior virologic failure on regimens containing these drugs. Additional inclusion criteria were age ≥ 18 years and patient willingness to switch. Demographic data, comorbidities (hyper-tension, type II diabetes mellitus, dyslipidemia), HIV-related events, prior antiretroviral regimens, and behavioral risk factors (e.g., smoking, alcohol use, risky sexual practices) were recorded.

### 2.2. Measurements

Two venous blood samples were obtained: one at baseline (immediately before switching to LAI) and one at 7 months.

The 7-month follow-up interval was chosen in accordance with current HIV treatment guidelines, which recommend monitoring lipid profiles every 3–12 months in people living with HIV, depending on individual cardiovascular risk [12,13]. This timeframe allowed sufficient duration for potential lipid changes to manifest after the switch to long-acting therapy, while maintaining feasibility and minimizing loss to follow-up.

The following variables were analyzed: lipid parameters (HDL, LDL, total cholesterol, triglycerides), HIV RNA viral load, and CD4+ T-cell count.

### 2.3. Sample Size Calculation

The target sample size was estimated at 31 participants (95% confidence, 8% precision) using Epidat software (v.4.2), based on the eligible population in our center; the final cohort included 39 patients.

### 2.4. Statistical Analysis

Quantitative variables were summarized as the mean ± standard deviation, and qualitative variables as frequencies with 95% confidence intervals. The Kolmogorov–Smirnov or Shapiro–Wilk tests were used to assess normality. Paired continuous variables were compared using repeated-measures ANOVA (parametric) or the Friedman test (non-parametric), and categorical variables using McNemar’s or Cochran’s Q tests. A mixed-effects model for repeated measures was employed to assess longitudinal changes. Statistical significance was set at *p* < 0.05. Analyses were performed using R Statistical Software (v4.1.2; R Core Team, 2021) and SPSS version 26 (IBM Corp., Armonk, NY, USA).

## 3. Results

The participants in this study had a mean age of 45.8 years, with a standard deviation of 12.8 years. Of these, 82.05% were men and 17.95% were women.

The average weight prior to starting long-acting injectable antiretroviral therapy was 78.6 kg, with a standard deviation of 14.8 kg.

Among the patients studied during the long-acting injectable antiretroviral therapy treatment period (*n* = 39 patients), one developed Kaposi’s sarcoma and another developed urethritis caused by Haemophilus influenzae.

Regarding previous antiretroviral therapy, 87.18% of patients had been receiving regimens based on integrase strand transfer inhibitors (INSTIs; primarily dolutegravir or bictegravir) or second-generation non-nucleoside reverse transcriptase inhibitors (NNRTIs; mainly rilpivirine) prior to switching to LAI. Two patients (5.13%) were receiving efavirenz-based therapy and two (5.13%) were on darunavir-based protease inhibitor regimens. As both efavirenz and protease inhibitors are associated with adverse lipid effects, their minimal representation in the cohort supports the overall metabolic neutrality observed after the switch to long-acting injectable therapy.

During the study period, no cardiovascular events occurred, no patient discontinued treatment for any reason, and no deaths were reported.

With respect to cardiovascular risk factors, 20.51% of the patients had arterial hypertension, 7.69% had type II diabetes mellitus, and 25.64% had dyslipidemia.

Among the patients with dyslipidemia, two were being treated with rosuvastatin, four with atorvastatin, two with pitavastatin, and two with simvastatin (Table 1).

Regarding HIV RNA (viral load) prior to the initiation of LAI, 7 patients (17.95%) had a viral load of <20 RNA copies/mL, and 32 patients (82.05%) had a viral load of 0 RNA copies/mL.

At 7 months after starting LAI, 3 patients (7.69%) had a viral load of <20 RNA copies/mL, 3 patients (7.69%) had a viral load of <40 RNA copies/mL, and 33 patients (84.61%) had a viral load of 0 RNA copies/mL.

Baseline—7-month comparison:Total cholesterol: No statistically significant differences were found when comparing the total cholesterol levels at the start and 7 months after the initiation of LAI, with a *p*-value of 0.17. However, there was a trend toward increased levels, with a baseline mean of 188.6 mg/dL (SD = 44 mg/dL), and after 7 months, a mean of 197.9 mg/dL (SD = 46.2 mg/dL).HDL cholesterol: Significant differences were found between the baseline and 7-month HDL levels, with a *p*-value of 0.00652. The baseline mean was 49.4 mg/dL (SD = 11.5 mg/dL), and the mean at 7 months was 53 mg/dL (SD = 11.9 mg/dL) Figure 1.LDL cholesterol: No statistically significant differences were found between LDL levels before starting the new therapy and after 7 months, with a *p*-value of 0.40. The baseline LDL mean was 115.4 mg/dL (SD = 35.5 mg/dL), and the mean at 7 months was 119.2 mg/dL (SD = 31.8 mg/dL).Triglycerides: No statistically significant differences were observed between triglyceride levels at the start of LAI and 7 months later, with a *p*-value of 0.80. The baseline mean triglyceride level was 132.6 mg/dL (SD = 109.4 mg/dL), and the mean at 7 months was 122.8 mg/dL (SD = 78 mg/dL).CD4: No statistically significant differences were detected in the total CD4 lymphocyte counts between baseline and 7 months after the initiation of long-acting injectable antiretroviral therapy (LAI) (*p* = 0.77). At the baseline, the mean CD4 count was 897.5 cells/mL (SD = 321.7), while the mean value at 7 months was 897.4 cells/mL (SD = 330.9). These findings indicate a remarkable stability of CD4 counts over the study period, suggesting that LAI maintained immunological status in this cohort without evidence of decline.

Distribution of HDL cholesterol levels (mg/dL) at the baseline and seven months after switching to long-acting injectable cabotegravir plus rilpivirine (*n* = 39). Violin plots with overlaid boxplots show the individual distributions; dashed lines connect paired values for each participant. Mean HDL levels increased from 49.4 mg/dL (SD = 11.5) at the baseline to 53.0 mg/dL (SD = 11.9) at seven months, representing a statistically significant increase (*p* = 0.0065).

## 4. Discussion

The findings of our study provide relevant insights into the metabolic consequences of switching from daily oral antiretroviral therapy (ART) to long-acting injectable cabotegravir plus rilpivirine (LAI), with a particular focus on lipid profiles. While randomized clinical trials such as FLAIR and ATLAS demonstrated that LAI is virologically non-inferior to oral regimens [1], evidence regarding its metabolic implications, especially concerning lipid metabolism, remains limited. Our results indicate that LAI maintains viral suppression and immunological stability without worsening total cholesterol, LDL cholesterol, or triglyceride levels, and with a significant improvement in HDL cholesterol. This pattern suggests that LAI has an overall favorable impact on lipid metabolism. This finding is of particular clinical importance, as people living with HIV (PLWH) carry an increased risk of cardiovascular disease (CVD) compared to the general population, attributable not only to traditional risk factors but also to chronic immune activation, systemic inflammation, and long-term exposure to antiretroviral therapy (ART) [1,2]. By demonstrating stability in atherogenic lipid fractions together with an increase in protective HDL, our data reinforce the evidence that LAI constitutes a metabolically safe and potentially beneficial alternative to conventional oral regimens.

The role of HDL in cardiovascular protection has been widely recognized, primarily due to its involvement in reverse cholesterol transport and its anti-inflammatory and antioxidative functions [5]. Although pharmacological interventions aimed at raising HDL levels have not consistently demonstrated reductions in cardiovascular events [7], observational studies indicate that higher HDL concentrations are associated with reduced cardiovascular risk in diverse populations, including PLWH [16]. Therefore, the significant increase in HDL observed in our cohort, even if modest in magnitude, represents a favorable metabolic change that may contribute to cardiovascular protection, particularly in patients already burdened with multiple cardiovascular risk factors. Importantly, this improvement mirrors findings reported in other real-world studies, such as the Japanese cohort described by Adachi et al., where HDL levels increased significantly following a switch to LAI [16]. The reproducibility of this outcome across different populations reinforces its validity and supports the notion that LAI may exert a beneficial influence on HDL metabolism in the context of long-acting antiretroviral therapy.

Beyond the clinical observation, several mechanisms may underlie the increase in HDL observed after switching to LAI. First, long-acting formulations minimize pharmacokinetic variability and reduce drug–drug metabolic interference compared with complex oral regimens, thereby stabilizing hepatic lipid processing [17,18]. Second, the improved adherence associated with injectable therapy ensures consistent antiretroviral exposure, avoiding the metabolic fluctuations that may arise from intermittent oral dosing [19]. Finally, sustained viral suppression and reduced immune activation under LAI could promote favorable lipid remodeling, as chronic inflammation and immune activation are known to impair HDL synthesis and function [20,21]. Together, these mechanisms may contribute to the observed improvement in HDL, suggesting that the metabolic benefits of LAI extend beyond simple lipid neutrality.

Another relevant aspect of our study is the observation that LAI did not worsen or adversely affect LDL, total cholesterol, or triglyceride levels after seven months, indicating maintenance of a stable lipid profile following the switch from oral therapy. This finding contrasts with the well-documented dyslipidemic effects of older ART classes, particularly protease inhibitors (PIs) and some nucleoside reverse transcriptase inhibitors (NRTIs), which are strongly associated with hypertriglyceridemia and elevated LDL cholesterol [7]. More contemporary regimens, such as those based on integrase strand transfer inhibitors (INSTIs) and newer non-nucleoside reverse transcriptase inhibitors (NNRTIs), have demonstrated more favorable lipid profiles in randomized and observational studies [12,13]. Nevertheless, concerns have emerged regarding the association of INSTIs, especially dolutegravir and bictegravir, with weight gain and potential metabolic complications [14]. In this context, the stability of total cholesterol, LDL, and triglyceride levels observed in our cohort further supports the metabolic safety of LAI, suggesting that this regimen maintains lipid homeostasis even in individuals previously exposed to INSTI- or NNRTI-based therapies.

The clinical implications of these findings extend beyond lipid metabolism alone. Dyslipidemia is a well-established risk factor for CVD, which remains one of the leading causes of morbidity and mortality among PLWH despite the widespread availability of effective ART [15]. Current cardiovascular risk prediction models, such as Framingham or SCORE, are not specifically validated in HIV populations, limiting their predictive accuracy. Nonetheless, sustained viral suppression and preservation of immune function have been associated with a lower incidence of dyslipidemia and reduced cardiovascular risk [2]. In our study, all patients maintained virological suppression, and CD4 counts remained stable, with no cases of profound immunosuppression (<350 cells/μL). These findings align with evidence from large cohorts, such as the RESPOND study, which reported that higher CD4 counts are associated with a reduced risk of metabolic complications, including dyslipidemia [2]. Therefore, the preserved immunovirological stability observed during LAI therapy likely contributed to maintaining a favorable and stable lipid profile, reinforcing the overall metabolic safety of long-acting injectable cabotegravir plus rilpivirine.

From a clinical perspective, the absence of deterioration in lipid parameters during LAI administration is especially relevant for patients with pre-existing metabolic comorbidities. As summarized in Table 2, approximately one-quarter of participants had dyslipidemia at the baseline, one-fifth had hypertension, and a subset were receiving statin therapy (rosuvastatin, atorvastatin, pitavastatin, or simvastatin). These patients continued their lipid-lowering regimens without modification throughout the study period, and no additional interventions were introduced. This continuity reinforces the interpretation that the favorable lipid outcomes observed were attributable to LAI rather than to therapeutic adjustments. While statin therapy and comorbidities such as hypertension and diabetes may have modulated individual lipid trajectories, the overall stability of lipid parameters across the cohort—including those not on statins—suggests that LAI neither interfered with nor attenuated the efficacy of concomitant lipid-lowering therapy. The potential modulatory role of these factors has been acknowledged in the statistical interpretation of results.

In addition, the sex distribution of our cohort was predominantly male (82.05%), which may limit the generalizability of our findings. Sex-related differences in lipid metabolism have been reported among people living with HIV, with women often exhibiting higher HDL cholesterol levels and distinct lipid responses to antiretroviral therapy, influenced by hormonal status and body fat distribution [4,22]. Therefore, future studies with a more balanced sex distribution are warranted to confirm whether the metabolic effects observed with LAI are consistent across genders.

Another dimension of our findings relates to the potential psychosocial and adherence benefits of LAI, which, although not the primary focus of our study, remain clinically relevant in the broader context of HIV management. Adherence to ART is the cornerstone of sustained virological suppression and prevention of drug resistance [23]. However, numerous factors—including stigma, fear of disclosure, daily pill burden, and treatment fatigue—may compromise adherence in real-world settings [24]. Long-acting regimens, by reducing the frequency of dosing and eliminating the need for daily oral administration, address several of these barriers simultaneously. While our analysis centered on lipid outcomes, the maintenance of viral suppression across our cohort underscores the likelihood that adherence remained high under LAI. This observation aligns with evidence from clinical trials such as ATLAS-2M, which demonstrated not only virological non-inferiority of LAI compared to oral therapy but also high levels of patient satisfaction and preference [25]. Although adherence itself does not directly modify lipid parameters, the ability of LAI to ensure consistent pharmacological exposure may indirectly promote metabolic stability and contribute to the maintenance of a favorable lipid profile, minimizing fluctuations associated with intermittent nonadherence.

The relevance of these observations is underscored by the fact that cardiovascular disease is increasingly recognized as a leading comorbidity among PLWH in the modern ART era [26]. Dyslipidemia, chronic inflammation, and immune activation converge to accelerate atherosclerotic processes in this population [27]. While ART has transformed HIV into a manageable chronic condition, the long-term toxicity of some regimens has amplified the prevalence of metabolic complications. Therefore, any new therapeutic approach must be evaluated not only for its virological efficacy but also for its cardiometabolic safety. In this regard, the absence of deterioration in LDL, total cholesterol, and triglyceride levels—together with the observed improvement in HDL—indicates that LAI is metabolically safe and may confer additional cardiovascular benefits. This balanced lipid response suggests that LAI occupies a particularly favorable position within the therapeutic landscape, especially for patients with elevated baseline cardiovascular risk. Moreover, as the HIV-positive population ages, with increasing numbers of patients over 50 years of age, the interplay between ART choice and cardiovascular risk will become an even more pressing clinical issue [28].

It is also worth considering the limitations of our study design in the interpretation of these findings. The relatively small sample size and the single-center nature of the study reduce the statistical power to detect subtle changes or rare adverse effects, and may limit generalizability to broader populations. The follow-up period of seven months, while sufficient to capture early metabolic trends, remains too short to fully evaluate long-term cardiovascular outcomes, which may take years to manifest clinically. Although the seven-month follow-up period provided valuable insight into early lipid adaptations following the switch to LAI, it remains too short to assess long-term cardiometabolic outcomes such as cardiovascular events or atherosclerotic progression. Therefore, our findings should be interpreted as reflecting short-term metabolic stability rather than definitive evidence of sustained cardioprotective effects. Nonetheless, our choice of a seven-month interval was consistent with the timing of routine follow-up assessments in clinical practice, allowing us to capture data that reflect real-world monitoring patterns. Furthermore, the limited sample size was dictated by the recent introduction of LAI at our institution, restricting the pool of eligible participants. Despite these constraints, the reproducibility of our results—showing stability in total cholesterol, LDL, and triglycerides, and an increase in HDL—together with their alignment with external reports, including the study by Adachi et al. [16], reinforces the robustness and external validity of our observations, supporting the overall metabolic safety of LAI.

In addition, the within-subject design of our study, in which each participant served as their own control, represents a methodological strength as it minimizes interindividual variability in metabolic parameters. However, the absence of an external parallel control group (i.e., individuals continuing oral ART) limits the ability to attribute lipid changes exclusively to the switch to long-acting injectable therapy. Consequently, causality cannot be definitively established within this design framework. Moreover, the relatively small sample size and single-center setting reduce statistical power and limit the generalizability of the findings, while potential selection bias cannot be completely excluded. These factors should be considered when interpreting the observed results.

Furthermore, the concomitant use of lipid-lowering therapy (e.g., statins) in a subset of participants represents a potential confounding factor that may have influenced lipid outcomes, particularly the observed increase in HDL cholesterol. Although these treatments remained unchanged throughout the study period, their possible contribution to the metabolic results cannot be fully excluded and should be considered when interpreting the observed lipid changes.

Another limitation concerns the absence of multivariate analyses to account for the potential influence of comorbidities or concomitant medications on lipid outcomes. While we initially considered stratified analyses by prior exposure to integrase strand transfer inhibitors (INSTIs) and NNRTIs, this was ultimately not feasible due to the very small proportion of patients (7.69%) who had not been exposed to either class. This limitation reflects the homogeneity of treatment history within our cohort but also underscores the need for larger studies capable of addressing such questions. Importantly, the baseline prevalence of cardiovascular comorbidities in our cohort—including hypertension (20.5%), type II diabetes mellitus (7.7%), and dyslipidemia (25.6%)—provides context for the interpretation of lipid outcomes. These comorbidities, along with the fact that several patients were already receiving statin therapy, highlight the clinical relevance of assessing metabolic safety in this population. While we cannot definitively exclude the influence of these factors, the consistent absence of lipid deterioration following LAI initiation—even in a comorbidity-burdened cohort—supports the conclusion that this regimen is metabolically safe and does not negatively impact lipid homeostasis in real-world conditions.

Finally, the implications of our findings extend to future research priorities. Longer-term prospective studies are needed to determine whether the HDL improvements observed here translate into measurable reductions in cardiovascular events. Subgroup analyses focusing on patients with pre-existing metabolic disorders, older age, or higher cardiovascular risk could help clarify whether the favorable effects of LAI are consistent across clinical strata or more pronounced in particular subgroups. Moreover, mechanistic studies exploring how long-acting cabotegravir and rilpivirine may influence lipid metabolism—either directly through pharmacological mechanisms or indirectly via improved adherence and reduced variability in drug exposure—could provide valuable insights. From a public health perspective, it will also be essential to evaluate the economic implications of LAI, as the reduced need for daily adherence and fewer clinic visits may offset the higher acquisition costs of injectable formulations [29]. Taken together, our findings establish a strong foundation for future research aimed at confirming the long-term cardiometabolic safety and potential lipid benefits of LAI in diverse populations.

When evaluating the clinical implications of our findings, it is essential to situate them within the broader literature on long-acting antiretroviral therapy. Clinical trials such as FLAIR and ATLAS have already demonstrated the non-inferiority of cabotegravir plus rilpivirine administered monthly compared to daily oral ART in maintaining viral suppression [21]. ATLAS-2M subsequently extended these observations, confirming similar outcomes with dosing every two months [29]. These data are critical because they provide the virological foundation that has enabled LAI to be implemented in clinical practice worldwide. Our study complements these pivotal trials by contributing real-world evidence specifically focused on lipid outcomes, which remain less frequently characterized in the context of LAI. The observation that LDL, total cholesterol, and triglyceride levels remained stable, while HDL significantly increased, reinforces the metabolic safety of LAI and suggests a potentially beneficial effect on cardiovascular risk modulation, even over the short-term.

These findings are consistent with emerging reports that highlight the relative metabolic neutrality of long-acting cabotegravir and rilpivirine compared with other antiretroviral regimens. For instance, Adachi et al. documented similar improvements in HDL without significant changes in LDL after switching to LAI [16], while other real-world cohorts have confirmed the absence of adverse metabolic effects with this regimen [7]. By aligning with these reports, our data further strengthen the growing body of evidence supporting the metabolic and cardiovascular safety of LAI. Although the field is still in its early stages, and longer follow-up is needed to determine whether these favorable lipid trends will persist and translate into reductions in clinical cardiovascular events, the current evidence consistently indicates that LAI does not worsen lipid profiles and may contribute to cardiovascular risk reduction over time.

Another important aspect to highlight is the role of baseline characteristics in interpreting lipid outcomes. Our cohort included patients with pre-existing comorbidities such as hypertension, type II diabetes, and dyslipidemia—conditions that independently contribute to cardiovascular risk. Moreover, a subset of patients was receiving statin therapy at the baseline, which could potentially mask subtle changes in lipid levels. Nevertheless, the consistent stability of lipid parameters across the cohort, regardless of statin use or comorbidities, reinforces that LAI did not adversely affect metabolic outcomes. The presence of these comorbidities in our cohort also enhances the external validity of our findings, as PLWH frequently present with multimorbidity, particularly as they age. Thus, our study reflects a real-world, clinically representative population, supporting the conclusion that LAI is a metabolically safe option even in individuals with underlying cardiovascular risk factors.

From a methodological standpoint, one of the challenges we encountered was the relatively limited statistical power due to the small sample size. While our analyses did not reveal significant differences in most lipid parameters, it remains possible that subtle effects went undetected. Additionally, we did not perform multivariate analyses due to the small number of patients unexposed to INSTIs or NNRTIs prior to switching, which limited the feasibility of stratified analyses. This limitation, however, underscores an opportunity for future multicenter studies with larger cohorts, where multivariate models could better elucidate the interplay between drug history, comorbidities, and lipid outcomes. Such work would help clarify whether the metabolic stability and HDL improvement observed in our study are generalizable across diverse populations and treatment backgrounds, further consolidating the evidence supporting the metabolic safety of LAI.

It is also worth reflecting on the temporal scope of our analysis. The seven-month follow-up was chosen to align with clinical practice patterns and provide an early assessment of metabolic changes after switching to LAI. While this timeframe offers valuable initial insights, a longer follow-up is necessary to determine the persistence of HDL improvements and evaluate the potential delayed effects on LDL or triglycerides. Cardiovascular outcomes, such as myocardial infarction or stroke, typically require years of follow-up and larger sample sizes for meaningful study. Therefore, our findings should be viewed as preliminary yet robust and encouraging evidence that LAI does not exacerbate metabolic risk and maintains lipid stability in the short-term. Prospective studies with extended follow-up and larger cohorts will be essential to confirm and expand upon these favorable metabolic outcomes.

Looking forward, future research should not only address lipid outcomes but also incorporate broader assessments of cardiovascular health, including inflammatory markers, endothelial function, and imaging studies of subclinical atherosclerosis. Such multidimensional analyses would provide a more comprehensive picture of the cardiovascular implications of LAI. Furthermore, exploring potential differential effects across subgroups—for example, older patients, women, or those with high baseline cardiovascular risk—could inform more personalized approaches to treatment. Given that cardiovascular disease is now one of the leading causes of morbidity and mortality among PLWH [26,27], understanding how LAI integrates into long-term cardiovascular risk management is a pressing clinical priority. Our findings suggest that LAI represents a metabolically safe therapeutic option that may fit well within strategies aimed at optimizing both virological control and cardiovascular health in people living with HIV.

Finally, the psychosocial dimension of LAI should not be overlooked. While our study focused primarily on lipid parameters, the high levels of viral suppression observed reinforce the notion that LAI supports strong adherence. Patient-reported outcomes from trials such as SOLAR and CUSTOMIZE have shown a clear preference for injectable regimens [30,31], with participants reporting reduced treatment fatigue, lower risk of missed doses, and decreased anxiety related to disclosure of HIV status. These factors, while not directly linked to lipid metabolism, indirectly contribute to the long-term success of therapy, including its favorable metabolic profile. A regimen that patients prefer and adhere to consistently is more likely to yield stable health outcomes over time. Thus, the combination of virological efficacy, metabolic safety, and high patient acceptability positions LAI as a promising and patient-centered advance in the continuum of HIV care.

## 5. Conclusions

In conclusion, our study demonstrates that switching from daily oral ART to long-acting Cabotegravir plus Rilpivirine is both virologically effective and metabolically safe over a seven-month follow-up period. No significant changes were observed in total cholesterol, LDL cholesterol, or triglycerides, confirming that LAI does not worsen the lipid profile. Importantly, a statistically significant increase in HDL cholesterol was identified, indicating a favorable metabolic effect that may contribute to lowering cardiovascular risk in people living with HIV.

These findings are consistent with those reported by Adachi and colleagues, who also documented an increase in HDL following the initiation of LAI, reinforcing the reproducibility and clinical relevance of this effect across different populations. Although the relatively small sample size and short follow-up duration limit the generalizability of our results, the consistency of these data provides valuable real-world evidence supporting the metabolic safety of LAI and its potential cardioprotective benefit.

Further multicenter studies with larger cohorts and extended follow-up are warranted to confirm these findings and more precisely define the long-term cardiometabolic implications of long-acting Cabotegravir plus Rilpivirine in diverse clinical settings.

## Figures and Tables

**Figure 1 microorganisms-14-00022-f001:**
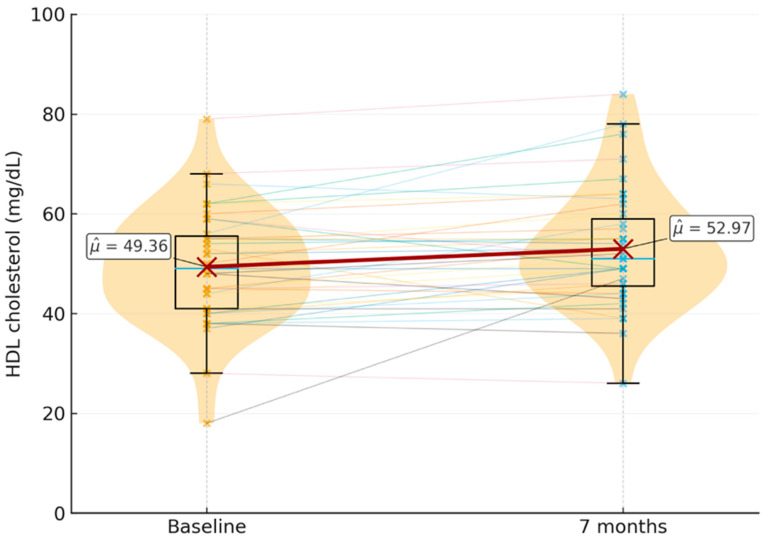
Changes in HDL cholesterol before and after switching to long-acting injectable therapy.

**Table 1 microorganisms-14-00022-t001:** Baseline demographic, clinical, and comorbidity characteristics of the study participants (*n* = 39).

Variable	Mean ± SD/*n* (%)	Details
Age (years)	45.8 ± 12.8	
Sex (male)	32 (82.1%)	
Weight (kg)	78.6 ± 14.8	
Hypertension	8 (20.51)	Controlled with ACE inhibitors or ARBs
Type II diabetes mellitus	3 (7.69)	All on oral hypoglycemics
Dyslipidemia	10 (25.64)	Rosuvastatin 2 (5.1%)
		Atorvastatin 4 (10.3%)
		Pitavastatin 2 (5.1%)
		Simvastatin 2 (5.1%)
Smoking	9 (23.08)	Current smokers
Alcohol use (moderate–regular)	6 (15.38)	

Abbreviations: ACE = angiotensin-converting enzyme; ARB = angiotensin II receptor blocker. Note: Percentages were calculated based on the total sample size (*n* = 39).

**Table 2 microorganisms-14-00022-t002:** Alterations in the metabolic profile before transitioning from oral to injectable therapy and seven months post transition.

Parameter	Before the Switch	Before the Switch	After 7 Months	After 7 Months	*p* Value
	Mean	SD	Mean	SD	
Total cholesterol (mg/dL)	188.6	44.0	197.9	46.2	0.17
LDL cholesterol (mg/dL)	115.4	35.5	119.2	31.8	0.40
HDL cholesterol (mg/dL)	49.4	11.5	53.0	11.9	**0.0065**
Triglycerides (mg/dL)	132.6	109.4	122.8	78.0	0.80

Note: Values are expressed as the mean ± standard deviation (SD). Statistically significant differences are highlighted in bold (*p* < 0.05).

## Data Availability

The data used in this study are existing, confidential healthcare records and are not publicly available due to privacy and ethical restrictions. Access to the data can be granted by the corresponding author upon reasonable request and with appropriate permissions.

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
