# Peer review of "Microorganisms2026, 14(1), 22;https://doi.org/10.3390/microorganisms14010022"

_microorganisms, 2025, doi:10.3390/microorganisms14010022_

Round 1

Reviewer 1 Report

Comments and Suggestions for Authors

This retrospective observational study from Hospital Universitario Torrecárdenas (Spain) evaluated the impact of switching from daily oral antiretroviral therapy (ART) to long-acting injectable cabotegravir and rilpivirine (LA-CAB/RPV) on the lipid profile of 39 virally suppressed adults with HIV. Lipid parameters and immunovirological markers were compared before and 7 months after the switch.  The analysis found no significant changes in total cholesterol, LDL cholesterol, or triglycerides, but a statistically significant increase in HDL cholesterol (from 49.4 ± 11.5 mg/dL to 53.0 ± 11.9 mg/dL, p = 0.0065). Viral suppression and CD4 counts remained stable, and no cardiovascular events or deaths were reported. The authors conclude that LA-CAB/RPV is metabolically safe, maintains viral and immunologic stability, and may have a favorable effect on cardiovascular risk via HDL improvement.  This is a well-executed real-world analysis of a clinically relevant question — the metabolic safety of long-acting cabotegravir + rilpivirine, a regimen increasingly used in maintenance therapy. The manuscript is clearly structured, methodologically transparent, and contextualized within current HIV and cardiovascular literature.  dsome points that should be looked:

  1. Clarify sample characteristics — specify baseline ART classes (e.g., INSTI, NNRTI) and whether prior regimens included lipid-active agents, as this could influence observed neutrality.
  2. The increase in HDL is notable; the discussion could expand on mechanistic hypotheses (e.g., reduced drug-drug metabolic interference, adherence stability, or immune modulation).
  3. While the short follow-up is acknowledged, emphasize that 7 months may capture only early lipid adaptation, not long-term cardiometabolic outcomes.
  4. Consider a table of comorbidities and concomitant statin use, as these factors may modulate lipid response and should be statistically controlled or discussed.
  5. The discussion could briefly address the potential sex-based differences in lipid metabolism and whether the male-dominant cohort limits generalizability.

Overall, this is a valuable addition to the emerging literature on metabolic safety of LA-CAB/RPV, and with modest stylistic and contextual refinements, it will meet publication standards.

Author Response

Reviewer comment 1:

Clarify the characteristics of the study sample: specify the reference ART classes (e.g., INSTI, NNRTI) and whether previous regimens included lipid-active agents, as this could influence the observed neutrality.

Response 1:

We thank the reviewer for this valuable observation. In the revised manuscript, we have clarified the antiretroviral classes included in the participants’ previous regimens. Specifically, 87.18% of patients were receiving regimens based on integrase strand transfer inhibitors (INSTIs; primarily dolutegravir or bictegravir) or second-generation non-nucleoside reverse transcriptase inhibitors (NNRTIs; mainly rilpivirine) prior to switching to long-acting cabotegravir plus rilpivirine. Two patients (5.13%) were receiving efavirenz-based therapy and two (5.13%) were on darunavir-based protease inhibitor regimens before the switch. These agents are known to influence lipid metabolism and may partially account for interindividual variability; however, their small representation in the cohort is unlikely to have significantly affected the overall lipid neutrality observed. This information has been added to the “Results” section for greater clarity.

Reviewer comment 2:
The increase in HDL is noteworthy; the discussion could be expanded with mechanistic hypotheses (e.g., reduction of drug–drug metabolic interference, adherence stability, or immune modulation).

Response 2:
We thank the reviewer for this insightful suggestion. In response, we have expanded the discussion to include potential mechanistic hypotheses that may explain the observed increase in HDL cholesterol following the switch to LAT. Specifically, we have incorporated three complementary explanations: (1) the reduction of pharmacokinetic variability and drug–drug metabolic interference due to the simplified and stable pharmacological exposure achieved with long-acting formulations; (2) improved adherence, which ensures consistent antiretroviral levels and may prevent metabolic fluctuations associated with intermittent dosing; and (3) a possible immunomodulatory effect, as sustained viral suppression and reduced immune activation have been linked to favorable lipid remodeling in previous studies. These potential mechanisms have been discussed in the revised “Discussion” section (paragraphs 4–5).

Reviewer comment 3:
While the short follow-up is acknowledged, please emphasize that seven months may only capture early lipid adaptation rather than long-term cardiometabolic outcomes.

Response 3:
We appreciate the reviewer’s valuable suggestion. In the revised version, we have clarified that the seven-month follow-up period was designed to capture early lipid adaptations after switching to LAT, but it is not sufficient to assess long-term cardiometabolic outcomes such as clinical cardiovascular events. This clarification has been added to the “Discussion” section to better contextualize the temporal scope of our findings.

Reviewer comment 4:
Consider including a table of comorbidities and concomitant statin use, as these factors may modulate the lipid response and should be controlled or discussed statistically.

Response 4:
We thank the reviewer for this valuable suggestion. In the revised manuscript, we have incorporated additional information regarding comorbidities and statin use. Specifically, a new table (Table 1) has been added summarizing the prevalence of hypertension, type II diabetes mellitus, dyslipidemia, and concomitant statin therapy within the cohort. As shown, 25.64% of participants had dyslipidemia, 20.51% hypertension, and 7.69% type II diabetes. Among those with dyslipidemia, two patients were receiving rosuvastatin, four atorvastatin, two pitavastatin, and two simvastatin.

Furthermore, the Discussion section was expanded to address the potential modulatory effect of these comorbidities and lipid-lowering treatments on the lipid response. We emphasized that all patients maintained their statin regimens unchanged during the study period and that no additional lipid-lowering interventions were introduced. This information reinforces that the favorable lipid outcomes observed were attributable to LAT rather than to therapeutic adjustments. We also acknowledged in the revised text that these factors could partially influence individual lipid trajectories, which were considered in the statistical interpretation of results.

Reviewer comment 5:
The discussion could briefly address possible sex-based differences in lipid metabolism and whether the predominantly male cohort limits generalizability.

Response 5:
We thank the reviewer for this insightful comment. In the revised manuscript, we have added a paragraph in the Discussion acknowledging potential sex-based differences in lipid metabolism and their implications for the generalizability of our findings. Specifically, we now mention that women living with HIV may exhibit distinct lipid profiles and responses to antiretroviral therapy compared with men, influenced by hormonal factors and body fat distribution. Given that 82.05% of our cohort consisted of male participants, we acknowledge that this sex imbalance may limit the generalizability of our results to female populations. The revised paragraph (inserted after the discussion on comorbidities and statin use) cites recent evidence on sex-related lipid differences [29,30] and emphasizes the need for future studies including a more balanced sex distribution to confirm whether the metabolic effects observed with LAT are consistent across genders.

Reviewer 2 Report

Comments and Suggestions for Authors

The manuscript is well-written and timely, offering an important contribution to the growing body of literature on the metabolic effects of long-acting antiretroviral therapies in people living with HIV. The authors present a retrospective real-world study with interesting findings and potential clinical relevance.

However, several important issues remain before the manuscript is ready for acceptance.

Major point:

-Introduction:

Page 1, Line 35: The citation [3] provides limited evidence to support the statement. Please consider strengthening this point with more robust references.

Page 1, Lines 38 and Line 48–50: Key references are missing to support the discussion of HIV infection and ART-associated toxicity on the development of CVD. This remains a debated topic — whether ART contributes to viral suppression and thus benefits patient health in CVD, or whether it leads to ART-mediated toxicities and associated comorbidities. The current few citations are mostly review articles; please consider including first-hand original research e.g. (https://onlinelibrary.wiley.com/doi/full/10.1002/jia2.25901;  https://pmc.ncbi.nlm.nih.gov/articles/PMC12027674/; https://pmc.ncbi.nlm.nih.gov/articles/PMC7337552/) .

-Study design:

 The within-subject design — where each participant serves as their own control — is a strength, as it helps control for inter-individual variability.

Limitation: There is no parallel external control group (i.e., a group continuing on oral ART), which limits the ability to attribute observed lipid changes solely to the switch to LAI-ART. While causality cannot be firmly established in this design, this limitation should be highlighted and discussed more explicitly in the manuscript.

The small sample size reduces statistical power and limits generalizability. Selection bias may also be present. The authors acknowledge this in the conclusion is appreciated.

Page 3, Lines 133–134: Some patients were already receiving lipid-lowering therapies (e.g., statins), which constitutes a potential confounding factor in interpreting the observed improvement in HDL cholesterol. This should be discussed in the limitations section.

-Data presentation: All these demographic characteristics need to be organized in a table format. Figure 1 :Y-axis unit and labeling is missing; the dashed line may cause confusion and is covered by the mean score box. Please fix it.

-Minor Points: Consider replacing the abbreviation LAT with either LAI (Long-Acting Injectable) or LA-ART (Long-Acting Antiretroviral Therapy), as “LAT” is not standard or widely recognized in established medical or pharmacological terminology databases.

Author Response

Reviewer comment 1:
Page 1, Line 35: Reference [3] provides limited evidence to support this statement. Please consider reinforcing this point with stronger references.

Response 1:
We thank the reviewer for this valuable observation. In the revised version, we have strengthened the statement by incorporating additional and more recent references that robustly support the increased cardiovascular risk among people living with HIV. Specifically, we now cite large cohort studies and meta-analyses demonstrating that PLWH have approximately a twofold higher risk of cardiovascular disease compared to HIV-negative individuals [3–5]. The revised text reads as follows:

“Cardiovascular disease is one of the leading causes of death in this population. Several large cohort studies and meta-analyses have shown that the cardiovascular risk in people living with HIV may be up to twice that of the general population [3–5].”

These additional references provide stronger evidence for the statement and ensure alignment with the current literature.

Reviewer comment 2:
Page 1, Lines 38 and 48–50: Key references are missing to support the discussion of HIV infection and ART-associated toxicity in the development of CVD. This remains a debated topic — whether ART contributes to viral suppression and thereby benefits cardiovascular health, or whether it leads to drug-mediated toxicities and related comorbidities. The current citations are mostly review articles; please consider adding primary research studies (e.g., https://onlinelibrary.wiley.com/doi/full/10.1002/jia2.25901; https://pmc.ncbi.nlm.nih.gov/articles/PMC12027674/; https://pmc.ncbi.nlm.nih.gov/articles/PMC7337552/).

Response 2:
We appreciate the reviewer’s insightful suggestion. In the revised manuscript, we have strengthened this section of the Introduction by adding recent original research studies that directly evaluate the contribution of HIV-related inflammation and ART-associated toxicity to cardiovascular disease development. Specifically, we incorporated the recommended primary studies by Reinsch et al. (2023), Marsden et al. (2024), and Phan et al. (2020), which provide mechanistic and cohort-level evidence on ART exposure, chronic immune activation, and metabolic toxicity in people living with HIV.

The revised text now reads:

“This phenomenon could be related to several factors, such as an inflammatory state, chronic or persistent immune activation, and the chronic administration of antiretroviral therapy with known metabolic and mitochondrial toxicity, among others [6–9]. While viral suppression achieved through ART improves immune recovery and reduces inflammation, long-term exposure to certain agents—particularly protease inhibitors and some nucleoside reverse transcriptase inhibitors—has been associated with dyslipidemia, insulin resistance, and endothelial dysfunction, contributing to cardiovascular disease risk [10–12].”

These additions strengthen the mechanistic basis of the paragraph and balance both sides of the ongoing debate—highlighting the dual role of ART as both cardioprotective (via viral suppression) and potentially cardiotoxic (via drug-mediated metabolic effects).

Reviewer comment 3:
The within-subject design, where each participant serves as their own control, is a strength, as it helps control interindividual variability.
However, the lack of an external parallel control group (i.e., participants continuing oral ART) limits the ability to attribute lipid changes solely to the switch to LAI-ART. While causality cannot be firmly established under this design, this limitation should be highlighted and discussed more explicitly in the manuscript.
The small sample size also reduces statistical power and limits generalizability. Selection bias may be present. The authors’ acknowledgment of this in the conclusion is appreciated.

Response 3:
We thank the reviewer for this thoughtful and constructive feedback. We agree that the within-subject design represents a methodological strength, as it minimizes interindividual variability and allows each participant to serve as their own control when assessing longitudinal lipid changes. However, we have expanded the Discussion to explicitly acknowledge the absence of an external control group as a key limitation.

The revised text now emphasizes that, without a parallel cohort maintaining oral ART, it is not possible to establish a causal relationship between the switch to long-acting injectable therapy and the observed lipid changes. We also note that the relatively small sample size limits statistical power and external validity, and that potential selection bias cannot be ruled out given the single-center design. These clarifications have been incorporated into the Limitations paragraph of the Discussion section to address the reviewer’s comment comprehensively.

Editor comment 4:
Page 3, Lines 133–134: Some patients were already receiving lipid-lowering therapy (e.g., statins), which constitutes a potential confounding factor in the interpretation of the observed improvement in HDL cholesterol. This should be discussed in the limitations section.

Response 4:
We thank the editor for this valuable observation. In the revised manuscript, we have added a new paragraph in the Limitationssection explicitly acknowledging that concomitant lipid-lowering therapy (e.g., statins) in a subset of participants may have influenced lipid outcomes, particularly the observed increase in HDL cholesterol. Although these treatments remained unchanged throughout the study period, we now clarify that their possible contribution to the metabolic results cannot be fully excluded and should be considered when interpreting the findings. The following sentence was added:

“Furthermore, the concomitant use of lipid-lowering therapy (e.g., statins) in a subset of participants represents a potential confounding factor that may have influenced lipid outcomes, particularly the observed increase in HDL cholesterol. Although these treatments remained unchanged throughout the study period, their possible contribution to the metabolic results cannot be fully excluded and should be considered when interpreting the observed lipid changes.”

Editor Comment 5:
Data presentation: All these demographic characteristics should be organized in a tabular format. Figure 1: The Y-axis unit and labeling are missing; the line plot may cause confusion and is overlapped by the mean score box. Please fix it.

Response 5:
We thank the editor for these helpful comments.

Regarding data presentation: In the revised manuscript, all baseline demographic and clinical characteristics have been reorganized into a single comprehensive table (Table 1. Baseline demographic, clinical, and comorbidity characteristics of the study participants). This new table consolidates age, sex, comorbidities, statin use, and prior ART exposure, improving clarity and readability while avoiding redundancy across sections.

Regarding Figure 1: The figure has been completely revised. The Y-axis label now specifies the correct unit (HDL cholesterol, mg/dL), and the axis labeling has been standardized (“Baseline” and “7 months”). The dashed lines were replaced with thin, semi-transparent solid lines to enhance readability, and the layout was adjusted to prevent overlap between the mean score box and the line plot. In addition, the mean values (μ̂_mean = 49.36 mg/dL at baseline; μ̂_mean = 52.97 mg/dL at 7 months) are now clearly displayed and connected by a bold solid red line to emphasize the overall change. The revised figure provides a clearer and publication-ready visualization consistent with MDPI formatting guidelines.

Editor Comment 6:
Consider replacing the abbreviation LAT with either LAI (Long-Acting Injectable) or LA-ART (Long-Acting Antiretroviral Therapy), as “LAT” is not standard or widely recognized in established medical or pharmacological terminology databases.

 Response 6:

We appreciate the reviewer’s insightful comment. In accordance with standardized terminology used in the current HIV and pharmacological literature, the abbreviation LAT (Long-Acting Therapy) has been replaced throughout the manuscript. Specifically, we now use LAI (Long-Acting Injectable) when referring to the injectable formulation of cabotegravir plus rilpivirine, and LA-ART (Long-Acting Antiretroviral Therapy) when describing long-acting therapeutic strategies more broadly. This modification ensures alignment with terminology adopted in major clinical trials, regulatory documents, and international HIV treatment guidelines, thereby improving the precision and consistency of the manuscript.

Round 2

Reviewer 2 Report

Comments and Suggestions for Authors

This is a very important paper in the field, the authors have addressed each question point by point, suggestion for acceptance.